# Atmospheric health burden across the century and the accelerating impact of temperature compared to pollution

Andrea Pozzer [1,2] ✉, Brendan Steffens [1], Yiannis Proestos [2], Jean Sciare[2], Dimitris Akritidis [1,3], Sourangsu Chowdhury [4], Katrin Burkart [5] & Sara Bacer[1]

Anthropogenic emissions alter atmospheric composition and therefore the climate, with implications for air pollution- and climate-related human health. Mortality attributable to air pollution and non-optimal temperature is a major concern, expected to shift under future climate change and socioeconomic scenarios. In this work, results from numerical simulations are used to assess future changes in mortality attributable to long-term exposure to both non-optimal temperature and air pollution simultaneously. Here we show that under a realistic scenario, end-of-century mortality could quadruple from present-day values to around 30 (95% confidence level:12-53) million people/ year. While pollution-related mortality is projected to increase five-fold, temperature-related mortality will experience a seven-fold rise, making it a more important health risk factor than air pollution for at least 20% of the world's population. These findings highlight the urgent need to implement stronger climate policies to prevent future loss of life, outweighing the benefits of air quality improvements alone.

Atmospheric conditions have a significant impact on human health: exposure to temperature extremes and to air pollution poses important risk for health. Increased air pollution has been associated directly with increased mortality[1]. Similarly, non-optimal temperature, defined as exposure to both high and low temperatures, has also been implicated with increased mortality[2]. The Global Burden of Diseases (GBD) estimate[3] finds that both these risk factors are responsible, in total, for the deaths of roughly 6.5 (95% confidence level: 5.3-7.5) million people per year, ~69% of which were caused by ambient air pollution. The most important atmospheric pollutant is $PM_{2.5}$ (i.e. particulate matter with aerodynamic diameter less than 2.5 $\mu m$), which accounts for roughly 90% of the mortality attributable to air pollution[4]. $PM_{2.5}$ consists of fine particulate which, thanks to its size, can pass through the respiratory barrier and enter the circulatory system, causing cardiovascular disease and other complications[5]. Simultaneous investigation of projected air pollution and non-optimal temperature related mortality has been already partially investigated[6], although only at urban and regional levels [e.g., refs. 7,8].

Human activities exert a substantial influence on both risk factors. In fact, anthropogenic emissions into the atmosphere alter atmospheric composition and consequently the climate. It is therefore of primary importance to project the evolution of air quality and surface temperature into the near and far future. In the Coupled Model Intercomparison Project (CMIP6) simulations[9,10], a series of Earth System Models (ESMs, i.e. global models with the added capability to explicitly represent biogeochemical processes[11]) have performed numerical simulations of temperature and air quality for the present and up to the end of the century[12].

In this study, we estimate the current and future atmospheric health burden by calculating the global mortality attributable to long-

[1]Atmospheric Chemistry Department, Max Planck Institute for Chemistry, Hahn-Meitner weg, Mainz 55128, Germany. [2]Climate and Atmosphere Research Center, The Cyprus Institute, 20 Konstantinou Kavafi Street, Nicosia 2121, Cyprus. [3]Department of Meteorology and Climatology, School of Geology, Aristotle University of Thessaloniki, Thessaloniki 54124, Greece. [4]CICERO Center for International Climate Research, Oslo 0349, Norway. [5]Department of Health Metrics Sciences, University of Washington, 15th Ave NE, 3980, Seattle 98195 WA, USA. ✉e-mail: andrea.pozzer@mpic.de

term exposure to non-optimal temperature and fine particulate matter at global scale. For this goal, we use model results from three ESMs participating in CMIP6. The model results are combined with state-of-the-art exposure-response functions for non-optimal temperature and air pollution to estimate the total long-term atmospheric-related attributable mortality. The estimates are presented as means of twenty-year periods, from 1990 to the end of the century (with ten-year intervals), so as to better represent the long-term effects. Three future climate scenarios based on the Shared Socio-economic Pathways [[13], SSP] are used, namely the most probable SSP2-4.5 ("middle of the road") scenario, the sustainable scenario SSP1-2.6 ("Taking the Green Road"), and the very pessimistic and unlikely SSP5-8.5 ("taking the highway") scenario[14]. Only exposure to $PM_{2.5}$, the most harmful pollutant, is considered here. As different models have differing accuracy in representing present day estimates[15], the models are downscaled and bias corrected (see section Methods) to ensure their consistency. Here we present the ensemble means of mortality estimates based on the three ESMs, with the uncertainties representing the 95% confidence interval.

## Results

### Global and regional results

The mortality attributable to long-term exposure to non-optimal temperature and $PM_{2.5}$ for 2000 (i.e., 1990–2009 mean) and the end of the century (i.e. 2080-2099 mean) is presented in Table 1, for each of the 7 super regions defined by the GBD[3] (results for individual countries can be found in the Supplementary Tables 1–4)). For the SSP2-4.5 scenario, the results are summarized spatially in Fig. 1, while in the supplement the same are presented for scenarios SSP1-2.6 and SSP5-8.5 (Supplementary Figs. 2–5). For the year 2000, the long-term health burden of the atmosphere is 5.7 (3.1–9.0) million $yr^{-1}$, with 28% of it attributable to non-optimal temperature exposure (1.6 (0.6–2.8) million $yr^{-1}$) and the remaining 72% (i.e., 4.1 (2.5–6.2) million $yr^{-1}$) to air pollution exposure. Significant attributable mortality from both the risk factors is estimated in South and East Asia, which represent 44% of the world population, but bearing 59% of the global health burden with attributable mortality of 1.2 (0.63–1.86) and 2.31 (1.36–3.45) million $yr^{-1}$, respectively. The high mortality values in these regions are attributable to the large population as well as to the high pollution levels. In fact, in both regions, the premature mortality attributable to air pollution is roughly a factor of 5 (for South Asia) and 3 (for East Asia) larger than that attributable to non-optimal temperature. On the other hand, in the GBD defined High-Income region[3] (i.e. Australasia, Western Europe, High-Income North America and High-Income Asia Pacific) the mortality attributable to non-optimal temperature is larger than that attributable to $PM_{2.5}$, as listed in Table 1, highlighting (i) the success of the air pollution control policies in the High-Income region and (ii) the importance of non-optimal temperature to the overall health burden in these regions.

The estimates for the end of the century suggest a remarkable increase in global mortality. Figure 2a depicts the mortality attributable to both risk factors for the entire time period (2000-2090). In 2090 (i.e., 2080–2099 mean), the overall attributable mortality increases to 30.3 (13.6-53.1) million $yr^{-1}$ in the SSP2-4.5 scenario, with 36.9 (15.9–66.1) in a climate mitigation scenario SPP1-2.6, and reaching up to 44.3 (17.6–79.9) million $yr^{-1}$ under the more pessimistic SSP5-8.5 scenario. This corresponds to a global population-weighted increase of the attributable mortality (Supplementary Fig. 6) from 96 (52–153) deaths for every 100k individuals for the year 2000 to 337 (151–590), 509 (220–913) and 581 (231–1049) deaths for every 100k individuals for SPP2-4.5, SSP1-2.6 and SSP5-8.5, respectively. Clearly, the projected changes in attributable mortality due to non-optimal temperature and fine particulate are regionally different, with large differences especially over North America and Central-East Europe (see Fig. 1). While in these regions the mortality attributable to $PM_{2.5}$ is

predicted to decrease, the one due to non-optimal temperature increases in all regions and more prominently in the subtropics and the northern extratropics. Globally, over time, we find a larger relative (albeit smaller in absolute values) increase in attributable deaths from exposure to non-optimal temperature as compared to $PM_{2.5}$. Under all scenarios, South and East Asia still exhibit the largest total attributable mortality. Nevertheless, by the end of the century, a larger impact on mortality from non-optimal temperature differences is expected. Specifically, for the South and East Asia, mortality attributable to fine particulate matter is predicted to be only a factor of 3.4 and 1.5 times larger than that attributable to non-optimal temperature, respectively. In Central and Eastern Europe, as well as southern Latin America, the mortality linked to non-optimal temperature is anticipated to surpass that attributed to ambient $PM_{2.5}$ by the end of the century (see Table 1) in all scenarios. Importantly, in High-Income region, while present-day mortality rates from air pollution and non-optimal temperatures are comparable, future scenarios suggest that temperature-related mortality will increase to 3 to 7 times that of air pollution, depending on the scenario. These results underscore the importance of climate change for human well-being and society as a whole.

### Factors in mortality growth

Figure 2a depicts the total global health burden from the two risk factors across the century. The larger projected increase for the SSP5-8.5 scenario in attributable mortality compared to SSP1-2.6 and SSP2-4.5 is clearly marked, as well as the proportionally larger increase in the mortality attributable to non-optimal temperature exposure, compared to mortality due to air pollution. The factors influencing the attributable mortality are also presented in Fig. 2b–d. We find a consistent increase in population-weighted average exposure (Fig. 2b) to temperature globally, across the century, increasing from 19 °C in 2000 to roughly 22.0, 23.4 and 24.3 °C at the end of the century, for scenario SSP1-2.6, SSP2-4.5 and SSP5-8.5, respectively, in agreement with the climate change projections[9]. On the other side, global population-weighted exposure to fine particulate matter is predicted to peak in the decades 2010, 2020 and 2040 (for the SSP1-2.6, SSP2-4.5 and SSP5-8.5 scenarios, respectively). This peak is followed by a steep decrease, bringing the exposure to $PM_{2.5}$ at the end of the century to $\sim 31 \, \mu g m^{-3}$ (SSP2-4.5 and SSP5-8.5) and $\sim 25 \, \mu g m^{-3}$ (SSP1-2.6), i.e. lower than that at the beginning of the century ($\sim 34 \, \mu g m^{-3}$). The reason behind the increase in the global mortality attributable to air pollution, in spite of the decreasing pollution exposure across the latter half of the century, is to be identified in the development of the population itself throughout the century. The population is projected to grow in both scenarios (see Fig. 2c), reaching a maximum of $\sim 9.3$ billion in 2070 for SSP2-4.5 and $\sim 8.4$ billion in 2050 for SSP5-8.5 and SSP1-2.6 where it reduces significantly afterwards. Notably, the average age of the population is projected to increase steadily from the year 2010 onward (see Fig. 2d, and also Supplementary Fig. 7), beginning at 32 years old in 2010 and reaching up to 46, 56 and 55 years by the end of the century in the SSP2-4.5, SSP1-2.6 and SSP5-8.5 scenarios, respectively. The baseline mortality rates (see Section Methods) increases with the increase of age (i.e. larger for older age classes than younger one), representing the higher mortality risk at older ages. This strong aging of the population is therefore the primary driver of the mortality associated with both non-optimal temperature and pollution attributable mortality on a global scale. In a few regions where the population age is already higher than the global average at the beginning of the century (such as Central-Eastern Europe and high-income countries, see Table 1), the population aging is not as pronounced as for the global average. For such regions, the decrease in exposure to air pollution therefore translates directly to a decrease in the mortality attributable to air pollution, while the changes to the temperature exposure cause a constant increase in mortality. This is particularly important for the scenario SSP1-2.6 in which, despite the decrease in

**Table 1 | Total attributable mortality (in thousands) for each of the 7 super regions defined by the Global Burden of Disease for the beginning and the end of the century for different scenarios**

| region | Population (millions) | Non-optimal temperature | Air pollution |
|---|---|---|---|
| | Year 2000 | | |
| Central-Eastern Europe / Central Asia | 409.02 | 278(126–436) | 450(261–713) |
| High-Income | 932.28 | 376(178–580) | 316(154–568) |
| Latin America / Caribbean | 442.21 | 43(17–78) | 112(59–190) |
| North Africa / Middle East | 412.62 | 98(23–201) | 231(132–383) |
| South Asia | 1,298.62 | 195(7–462) | 964(627–1402) |
| Sub-Saharan Africa | 607.87 | 52(13–131) | 243(123–440) |
| Southeast-East Asia / Oceania | 1787.12 | 555(218–949) | 1751(1141–2501) |
| World | 5871.45 | 1593(581–2832) | 4061(2493–6188) |
| | Year 2090, SSP1-2.6 | | |
| Central-Eastern Europe / Central Asia | 293.8 | 1196 (481–1941) | 504 (254–1029) |
| High-Income | 1208.0 | 2126 (864–3462) | 303 (101–846) |
| Latin America / Caribbean | 467.54 | 499 (164–951) | 687 (308–1300) |
| North Africa / Middle East | 666.56 | 1875 (260–4084) | 2962 (1638–4918) |
| South Asia | 1666.21 | 2935 (39–7232) | 8674 (5276–13,338) |
| Sub-Saharan Africa | 1643.37 | 954 (137–2463) | 3314 (1733–5817) |
| Southeast-East Asia / Oceania | 1306.28 | 4946 (1529–8950) | 5934 (3164–9873) |
| WORLD | 7239.2 | 14,516 (3474–29,047) | 22,365 (12,470–37,092) |
| | Year 2090, SSP2-4.5 | | |
| Central-Eastern Europe / Central Asia | 346.23 | 872(342–1421) | 508(255–972) |
| High-Income | 1,194.44 | 1,752(672–2892) | 390(130–982) |
| Latin America / Caribbean | 602.86 | 392(105–765) | 712(352–1247) |
| North Africa / Middle East | 893.58 | 1,395(158–3081) | 2574(1449–4223) |
| South Asia | 2221.80 | 2,358(29–5740) | 7991(5138–11,737) |
| Sub-Saharan Africa | 2227.73 | 629(51–1669) | 2256(1176–3981) |
| Southeast-East Asia / Oceania | 1521.64 | 3,389(869-6299) | 5122(2879–8161) |
| World | 8995.74 | 10,777(2223-21,845) | 19,539(11,373-31,280) |
| | Year 2090, SSP5-8.5 | | |
| Central-Eastern Europe / Central Asia | 317.75 | 1365(447–2292) | 764(375–1434) |
| High-Income | 1685.75 | 2366(729–4076) | 857(332–1849) |
| Latin America / Caribbean | 434.92 | 587(69–1237) | 932(477–1580) |
| North Africa / Middle East | 655.12 | 1980(130–4527) | 3167(1782-5173) |
| South Asia | 1640.85 | 3952(22–9788) | 10,048(6358–14,955) |
| Sub-Saharan Africa | 1596.46 | 1495(32–3896) | 3623(1954–6184) |
| Southeast-East Asia / Oceania | 1302.61 | 5622(673–11,189) | 7575(4285–11,888) |
| World | 7614.3 | 17,336(2101–36,935) | 26,933(15,548–43,005) |

PM$_{2.5}$ exposure and the moderate increase in the non-optimal temperature exposure, the mortality attributable to both risk factors increases drastically through the century.

Most importantly, the non-optimal temperature attributable mortality is associated with a large increase in the warm temperature exposure, consistent with the climate change prediction in the examined models. The mortality due to warm temperature exposure increases by factors of 22, 23, and 52, for SSP2-4.5, SPP1-2.6 and SSP5-8.5, respectively, across the century. Mortality due to cold temperature, on the other hand, only increases by factors of 5, 7.5 and 6 for SSP2-4.5, SSP1-2.6 and SSP5-8.5, respectively, for the same time span (Supplementary Fig. 8). While mortality due to cold temperature exposure is reduced by the increasing global temperatures across the century, this reduction is countered and indeed reversed by population growth and the increase in average age across the century, as described before. It must be stressed that, depending on the location, climate change could increase the temperature variability[16], possibly even enhancing the frequency of cold days. The graphical distribution of non-optimal temperature attributable mortality from warm and cold temperatures exposure is depicted graphically in the Supplementary Fig. 9. The largest increase in warm temperature attributable mortality is especially pronounced in tropical regions, such as the Sub-Saharan Africa, South Asia, Indonesia and Central America, although increases are also present in extratopics, such as the Mediterranean area. This increase could have significant impacts, especially in the vulnerable Middle East and North Africa region, further exacerbating the societal implications of climate change, such as enhanced migration and deterioration of farmers' life quality[17].

## Discussion

In this work, the approach used follows that of the GBD as closely as possible, and our results are similar[3] for the period 2000–2019. For the temperature-related attributable mortality, the exposure-response functions developed by GBD[2] have been implemented. Daily cause-specific mortality data from 9 countries for a total of 64.9 million deaths, spanning 29% of the global population as well as approximately 95% of the inhabited global temperature range and 79% of socio-demographic conditions, were used to define the cause-specific

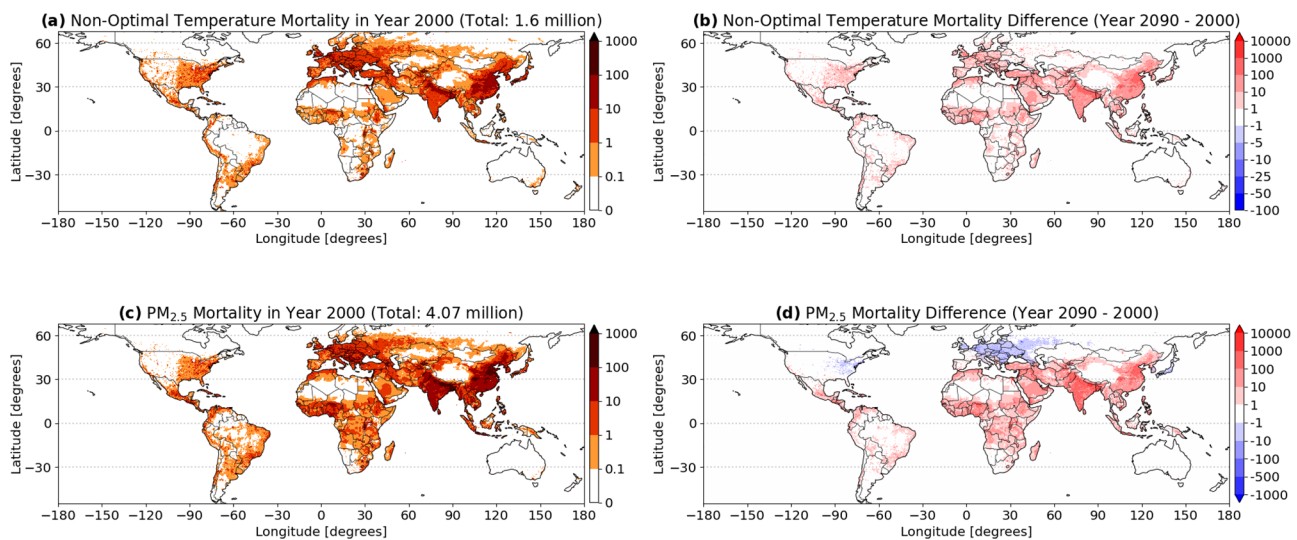

**Fig. 1 | Spatial distribution of the annual mortality.** Mortality attributable to long-term exposure to non-optimal temperature (**a**) and to air pollution (**c**) at the beginning of the century. The differences for the same results by the end of the century in the Shared Socio-economic Pathways SSP2-4.5 scenario are presented (**b**) and (**d**).

**Fig. 2 | Changes in the period 2000–2090 for the Shared Socio-economic Pathways scenarios SSP1-2.6 (dotted curves), SSP2-4.5 (solid curves) and SSP5-8.5 (dashed curves). a** Global mortality attributable to non-optimal temperature (red) and to air pollution (blue). **b** Global population weighted exposure to non-optimal temperature (red, right vertical axis) and air pollution (PM$_{2.5}$, blue, left vertical axis). **c** Global population. **d** Global population average age.

attributable burden. Furthermore, only the same-day effect is included (i.e., no lag period is assumed), which could result in an underestimation of the attributable mortality, especially for the cold temperature[2]. In another similar study[18], a non-optimal temperature attributable burden for all-cause mortality of ~5.0 (4.0–5.9) million deaths is estimated in the period 2000–2019, which is a factor of ~3 higher than the value presented here. The cause-specific association and the lack of lag period could explain the differences to our results. Furthermore, different exposure-response function is used than here, which is specific to the location and includes socio-economical information[19]. Finally, we did not considered that the impact of changes in non-optimal temperature is strongly affected by various modifiers such as income and working status, making population subgroups particularly vulnerable[20].

Different exposure-response functions are also available for pollution-related attributable mortality[4], producing different estimates. For example, an all-cause air pollution-attributable death total of 8.1 (5.9–9.9) million for the year 2019, a factor of 2 higher than the estimate obtained here, has been also calculated[21]. It must be stressed as well that we neglect the premature mortality attributable to long-term exposure to ozone, known to be an important risk for pulmonary diseases. To estimate the mortality attributable to ozone exposure, a specific ozone metric is necessary[22]. However, this metric can only be derived from hourly ozone values, which were not present in the CMIP6 database for the numerical simulations of the SSP2-4.5 scenario employed in this study. Nevertheless, ozone-attributable mortality is estimated to be roughly a factor of 10 lower than the one due to fine particulate[4].

On one side, based on these considerations and the comparison with other studies for the present day, there are therefore indications that the values estimated in this work are lower-end estimates. On the other side, we also assume a constant baseline mortality ratio, reflecting an unchanging situation in health care, nutrition, and other living conditions in the future. It has been shown that the use of projected baseline mortality ratios could decrease the mortality between ~20 and 50% in developed and developing regions, respectively, with a global average of ~40%[23]. This decrease could potentially offset the attributable mortality increase due to the population aging alone and indicates that our results could overestimate future attributable mortality. As the projections of baseline mortality rates are also affected by large uncertainties and no official predictions are present which have been produced simultaneously with the population projections, we decided to keep the present-day values constant also for the future, similarly to other works [e.g., refs. 24,25]. Nevertheless, our results are consistent with a previous study[23], which estimated the attributable mortality to $PM_{2.5}$ with one model and for the scenario SSP3-7.0 to be 6.83 (5.68–7.92) million $yr^{-1}$ for the period 2005–2014, with an increase to 25.7 (21.4–29.8) million $yr^{-1}$ by the end of the century.

The projected scenarios are investigating plausible futures. In this work we use model results from the SSP2-4.5, considered to be the most probable future, SSP1-2.6, a strong mitigation scenario, and the SSP5-8.5, a highly unlikely pessimistic scenario[14]. The latter presents stringent air pollution controls but no climate change policies, and therefore should be considered highly skewed towards non-optimal temperature mortality increase. On the other hand, scenario SSP1-2.6 contains very strong climate as well as air quality mitigation measures. In this scenario, the climate mitigation measures result also in additional co-benefits in terms of lower pollutant emissions[26], obtaining the lowest values of $PM_{2.5}$ at the end of the century.

Finally, we neglect any synergistic effects on cardiovascular disease due to simultaneous exposure to higher temperatures and higher concentrations of air pollutants, which can cause even higher health burdens than exposure to high temperature or high pollution alone[27]. Although a stronger relationship between $PM_{2.5}$ and health at higher temperatures is a possibility, there is currently insufficient evidence to include such effects.

The projected mortality attributable to long-term exposure to these atmospheric risk factors is expected to increase significantly in the future. Specifically, it is anticipated to rise from 5.7 (3.1–9.0) million $yr^{-1}$ in the year 2000 to 30.3 (13.6–53.1) million $yr^{-1}$ by the year 2090 under a realistic-medium scenario (SSP2-4.5), with non-optimal temperature exposure accounting for 36% of this total. Under a pessimistic scenario (SSP5-8.5), the predicted mortality attributable to these risk factors increases further to 44.3 (17.6–79.9) million $yr^{-1}$, with non-optimal temperature exposure accounting for 39% of this total. Finally, for a strong air pollution and climate mitigation scenario (SSP1-2.6), the total predicted mortality attributable reaches 36.9 (15.9-66.1) million $yr^{-1}$, with non-optimal temperature exposure accounting for 40% of this total. Independently on the scenario, the relative growth of non-optimal temperature is remarkable, as for the 2000 decade it accounts only for 28% of the total mortality attributable to these atmospheric risk factors. The analysis of the factors contributing to this increase reveals that population growth and aging are the primary drivers of this phenomenon. On a global scale, while mortality attributable to fine particulate matter is projected to increase by a factor of 5.5, 4.8 and 6.6 (for scenarios SSP2-4.5, SSP1-2.6 and SSP5-8.5, respectively), the mortality associated with non-optimal temperature exposure is expected to rise by a factor of 6.7, 9.1 and 10.9 (for scenarios SSP2-4.5, SSP1-2.6 and SSP5-8.5, respectively). Furthermore, it is noteworthy that the death toll from non-optimal temperature exposure already exceeds for present day that from air pollution in high-income countries. Projections indicate that this trend will extend to large regions such as Central and Eastern Europe, including dozens of populous countries worldwide (e.g., United States, Canada, Argentina, Chile, Algeria, Somalia, Germany, France, Australia, New Zealand, Ukraine, Russia, Japan, and South Korea). It is anticipated that climate factors, and specifically non-optimal temperature exposure, will become a more significant contributor to the overall health burden than air quality at the end of the century for roughly 21% (SSP2-4.5), 30% (SSP1-2.6), 32% (SSP5-8.5) of the world's population, in contrast to 14% for the year 2000. This underscores that climate change poses a direct threat to human life expectancy and could emerge as a major determinant of global health burdens, potentially offsetting the mortality reductions achieved through air pollution control policies.

## Methods
The attributable mortality to non-optimal temperature and fine particulate is estimated in each location (i.e., in each model grid-box) using the equation (1).

$$\text{Mort}_{d,X,\Delta a} = \text{BMR}_{d,\Delta a} \cdot \text{Pop}_{\Delta a} \cdot \text{AF}_{d,X,\Delta a} \tag{1}$$

where $\Delta a$ represents the age group, $d$ the disease, $X$ the risk factor (i.e., temperature or $PM_{2.5}$), AF the attributable fraction (i.e., the fraction of deaths due to $d$ and attributable to $X$), Pop the population, and BMR the cause-specific baseline mortality rate (i.e., the fraction of deaths due to $d$).

### Temperature and $PM_{2.5}$
We employ output data from global numerical simulations conducted within the CMIP6[9]. We select models that provide both daily near-surface temperature and monthly surface $PM_{2.5}$ mixing ratio (then converted into annual average mass concentration) for the historical period (1980–2014) and three climate warming scenarios, SSP1-2.6, SSP2-4.5 and SSP5-8.5 (2015–2099). Moreover, to ensure accuracy and consistency, these models must have a nominal horizontal resolution of 100 km. Our analysis is therefore limited to three global models: CESM2-WACCM[28,29], GFDL-ESM4[30,31], and MRI-ESM2-0[32,33], which are the only ones with all the described requirements. We assume that each model contributes equally to our analysis.

Both fields, near-surface temperature and $PM_{2.5}$ concentration, were downscaled and bias-corrected in this work. To develop the bias-adjusted (daily mean) field of near-surface temperature, we initially apply the Climate Imprint algorithm. This involves calibrating and downscaling each model over the baseline (historical) period of 1980–2014 against the proxy observational WFDE5 (WATCH Forcing Data ERA5) reanalysis[34] data set with a spatial resolution of 0.5 × 0.5 degrees. Next, we implement the climate-signal preserving, Quantile Delta Mapping algorithm using the calibrated/downscaled data sets (1980–2014) and the SSP-based future projections (2015–2100) to obtain the final bias-adjusted and statistically disaggregated field for each selected model and climate warming scenario[35]. Both bias-adjustment algorithms mentioned above are sourced from the Pacific Climate Impacts Consortium's Climate Downscaling (ClimDown) package available in the R statistical programming language[36,37].

For the $PM_{2.5}$ concentration data, the bilinear interpolation and the delta method are applied for the downscaling and bias correction, respectively. The reference $PM_{2.5}$ concentration field is the observational data set of[38], which is based on a combination of satellite observations and model results, calibrated using ground-based observations incorporated with a Geographically Weighted Regression (GWR). In this study, the annual mean global GWR-adjusted $PM_{2.5}$ estimates at the resolution of 0.1 × 0.1 degrees are used [v5.GL.02,[38]]. In order to consider $PM_{2.5}$ concentration climatologies, the bias correction is applied on temporal means of 20 years[39], considered long enough to show future changes according to the CMIP6[40]. Thus, the following computations are performed as equation (2) and (3).

$$PM_{2.5, m, t_i}(x, y) = \overline{PM}_{2.5, obs, 1998-2017}(x, y) \cdot \Delta_{m, t_i}(x, y) \qquad (2)$$

$$\text{where } \Delta_{m, t_i}(x, y) = \left( \frac{\overline{PM}_{2.5, m, t_i}}{\overline{PM}_{2.5, m, baseline}} \right) \qquad (3)$$

where the bar denotes a temporal mean, $(x, y)$ is the location (longitude, latitude) of the data, $m$ refers to the selected model, baseline indicates the past 20-year period (i.e., 1990–2009), $t_i$ are the time slices of 20 years with step of 10 years, i.e., 1990–2009 to be representative for the year 2000, 2000–2019 for the year 2010, ... 2080–2099 for the year 2090. For consistency, we consider 20 years also for the observational data set (1998–2017), although the available period is longer (1998–2020). To be noted that $\Delta_{m, t_i}(x, y)$, computed with model data, is downscaled to the resolution of the observations. Thus, the final bias-corrected $PM_{2.5}$ concentration has the resolution of 0.1 × 0.1 degrees.

In order to perform a consistent comparison between the results obtained for the two risk factors, daily bias-adjusted temperature estimates are also averaged over 20-year periods, every 10 years. Since mortality attributable to temperature is determined daily, multi-annual daily means are computed this time for the same time slices.

Finally, both temperature and $PM_{2.5}$ concentration are regridded to the target grid of our analysis: the grid of the population data set at the resolution of 7.5 arc-min, i.e. 0.125 × 0.125 degrees.

### Population and base mortality rate

We use gridded population for the base year 2000 and gridded population projections at ten-year intervals for 2010–2090 at the resolution of one-eighth degree (7.5 arc-minutes) available from SEDAC (Socio-Economic Data and Application Center, https://sedac.ciesin.columbia.edu/data/collection/ssp). The projections are consistent with the SSPs[41].

In order to distribute population data by age, we use the information on the number of individuals per age group provided by IIASA (International Institute for Applied Systems Analysis, https://tntcat.iiasa.ac.at/SspDb/dsd?Action=htmlpage&page=about.). This information is

available from 2010 until 2100 every five years at the country level. We use data every ten years (from 2010 to 2090) and assume that distribution in 2010 is the same as in 2000. Age-distributed population is obtained by multiplying the ratio of the population belonging to that age group at the country level (derived from IIASA data) by the population data (from SEDAC).

Caused-specific baseline mortality rate (BMR) is downloaded from the Institute for Health Metrics and Evaluation (IHME, https://vizhub.healthdata.org/gbd-results/) for the years 1990–2009, for the diseases and age groups considered in this study. Attributable mortality (equation (1)) is estimated by applying the 20-year mean of BMRs, which is kept constant also in the future estimation.

### Exposure-Response Functions and Attributable Fraction

Attributable fractions are derived from the so-called exposure-response functions (ERFs), which are based on the relative risks (RRs) established by different studies. The non-optimal temperature ERF are established on vital registration data (i.e. death certificates) while the fine particulate ERF are based on individual cohort studies metaregressed. The risk model coefficients defining the ERFs are adjusted and updated as soon as more data are available so that the *RRs* estimated from the ERFs are close to the *RRs* defined by the cohort epidemiological studies. More precisely, $AF = (RR - 1)/RR$, where RR is estimated from the considered ERFs; there are in fact several functions for $PM_{2.5}$[4] and just a few for temperature[3,19]. In this study, we use the ERFs from the GBD (GBD2019) both for non-optimal temperature[2] and for fine particulate[3] exposure.

The ERFs for non-optimal temperature are derived with the meta-regression–Bayesian, regularised, trimmed (MR-BRT) tool. They are defined for the entire population, without distinction by age, for 17 causes of deaths: external causes (i.e. injuries), non-external causes (i.e. diseases), and metabolic diseases (i.e. diabetes and chronic kidney disease). Among these, we estimate the attributable mortality associated with non-external causes and metabolic diseases: cardiomyopathy and myocarditis (CMP), hypertensive heart disease (HTN), ischemic heart disease (IHD), stroke, lower respiratory infections (LRI), chronic obstructive pulmonary disease (COPD), chronic kidney disease (CKD) and diabetes. Each cause-specific ERF is differentiated by climate zone, identified from the temporal mean (between 1980 and 2016) of daily mean temperatures in that location; 23 climate zones are found for populated areas, from 6 °C to 28 °C. Like in ref. [2], we consider here 23 climate zones (6 °C–28 °C) based on the temporal mean of (multi-annual) daily mean temperatures for each 20-year period. We compute daily *AFs* in each grid-box by using the multi-annual daily mean temperature and the cause-specific ERF of the climate zone to which that grid-box belongs to.

The ERFs for $PM_{2.5}$ are splines generated with the MR-BRT tool and input data from epidemiologic studies of exposure to ambient air pollution, household air pollution from the use of solid fuels, and secondhand tobacco smoke from the GBD 2019[3]. These ERFs are cause-specific for IHD, stroke, COPD, lung cancer (LC), and diabetes mellitus type 2 (T2 DM) for people of age 25+ and for LRI for children of 0–5 years; the ERFs for cardiovascular causes (i.e., IHD and stroke) are differentiated into age groups of 5 years. In the case of $PM_{2.5}$ exposure, the ERFs have a global validity (they do not depend on climate zones) and are based on annual mean concentrations, therefore, we compute annual AFs in each grid-box using the cause-specific ERFs and the (20-year) mean $PM_{2.5}$ concentration of that grid-box. Following the GBD2019 approach, the theoretical minimum-risk exposure level is obtained from an uniform distribution between 2.4 and 5.9 $\mu g\,m^{-3}$.

### Mortality estimates

In order to perform the computation of equation (1) in each model grid-box, the data sets previously described were post processed, so to be on the same grid. The target grid is the one of the population

(7.5 arc-min, i.e., 0.125 × 0.125 degrees). We use the national identifier grid at the resolution of 2.5 arc-min from SEDAC, to convert the data at the country level (age groups and *BMRs*) to gridded data.

While equation (1) can directly be used to compute annual mortality attributable to long-term exposure to PM$_{2.5}$, annual mortality attributable to non-optimal temperature is computed with equation (4).

$$\text{Mort}_{d, X = temp, \Delta a} = \frac{1}{365} \cdot \text{BMR}_{d, \Delta a} \cdot \text{Pop}_{\Delta a} \sum_{day=1}^{365} \text{AF}_{d, X = temp, \Delta a}^{(day)} \qquad (4)$$

where $\text{AF}_{d, X = temp, \Delta a}^{(day)}$ are daily AFs.

Therefore, with equations (1) and (4) we compute mortality estimates for different diseases and age groups. It must be noted that mortality estimate is computed for all $\Delta a$ even when *AF* is not dependent on age (in this case, the same *AF* is used for all age groups). The sum of $Mort_{d,X,\Delta a}$ for all considered diseases and age groups gives the total number of deaths attributable to non-optimal temperature or fine particulate matter.

The final results have the resolution of 0.125 × 0.125 degrees and extend between −54.95 S and 67.95 N (which is the largest latitudinal extension common to all data sets; this corresponds to the observational data set of PM$_{2.5}$). The confidence level of mortality estimates is computed by using the confidence level intervals of *RRs* provided by GBD2019[2,3] in equations (1) and (4).

We also compute the relative contributions to mortality attributable to non-optimal temperature for cold temperature exposure and warm temperature exposure, specifically. This is done by computing the minimum of each ERF, described earlier, for each disease and each climate zone. The temperature associated with these minima is the so-called theoretical minimum-risk exposure level (TMREL) or minimum-mortality temperature (MMT)[2]. TMRELs are location and year-specific, although in this work were kept fixed and based on the climate zone. As we evaluate each ERF during the mortality calculation for a given grid cell of temperature T, we track whether T ≥ MMT or T < MMT, and associate the calculated mortality with warm temperature exposure or cold temperature exposure, respectively.

### Reporting summary
Further information on research design is available in the Nature Portfolio Reporting Summary linked to this article.

## Data availability
All data used in this work are publicly available. The results are presented in tabulated format in the supplementary information of this manuscript. The processed data with total mortality, generated in this study and used to produce the figures and the tables, is available at https://doi.org/10.17617/3.ITPOI5.

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

## Acknowledgements

We acknowledge the contribution of Jeffrey Stanaway and Michael Brauer to the Global Burden of Disease Study (GBD), whose data and exposure-response functions have been extensively used in this work. B.S. was supported by the European Union's Horizon Europe research and innovation program under Grant Agreement No 101057131, Climate Action To Advance HeaLthY Societies in Europe (CATALYSE). S.B. was supported by the postdoctoral fellowship of the Alexander von Humboldt Foundation.

## Author contributions

A.P. and S.B. designed the study. S.B. and A.P. and B.S. performed the calculations with the help of D.A. B.S. prepared the tables and the figures. A.P. wrote the manuscript with the help of S.B., S.C., and B.S. Data was provided by Y.P., J.S., and K.B.

## Funding

## Competing interests

The authors declare no competing interests.
