## [Peer Review File · Nature Communications]

REVIEWER COMMENTS

Reviewer #1 (Remarks to the Author):

General

- Very cool and interesting use of the most recent GBD data techniques and novel risk measures for impact assessment! Methods seem reproducible and in line with current climate modelling norms.

- Be careful with the interpretation here when stating ‘non-optimal temperature as more important’ as built into the SSPs are measures that reduce air pollution, whilst built into the scenarios there is an assumption around temperature increase. Probably reducing climate change will also improve air quality. SSP5-8.5 is unlikely to occur and is more the ‘worst case scenario’.

- In the body of the text you highlight the importance of air pollution policy but the title of the piece makes it seem that temperature and air pollution are pitted against each other. At the moment air pollution is a bigger share of burden of disease relative to temperature, thus I think there can also be a different way to frame the issue. I think that titling it “Atmospheric Health Burden Across the Century: the Accelerating Impact of Temperature “ is still a quite effective title. Also I think it’s a bit of a unfair comparison if I understand things correctly because the temperature mortality relationships are not taking into consideration adaptation measures while the PM2.5 calculations are, since SSP5 has stringent air pollution control and SSP2 has intermediate air pollution control. To demonstrate combined mitigation and air pollution benefits maybe an analysis using SSP1-2.6 could be used? That could reinforce your conclusion that effective climate change policies are also important

Abstract

- The concluding statement regarding ‘simultaneously implementing more effective climate change policies’ is very nice. I agree but not sure if the analysis demonstrates this since there is no explicit comparison of air pollution control scenarios to mitigation control scenarios.

- Check whether or not British English or American English is used for the word ‘aging/ageing’ in Nature/the editor.

- I think that the phrase including the word ‘potentially’ for SSP5-8.5 should be rewritten into something more moderate.

- Specify briefly how this work either adds onto or reinforces other scientific findings if word limit permits. E.g. It is probably one of the few analyzing temperature and air pollution simultaneously at the global level , or that air quality improvements may be offset by non-optimal temperatures

Introduction

- Be more specific in general when making broad claims and statements for example about 'non-optimal temperatures' as there are varying definitions which in some papers vary based on location in order to control slightly for local acclimatization/adaptation (e.g. Gasspirinis 'Projections of temperature-related excess mortality under climate change scenarios' 2017 article)

- Choose one way to frame the 'exposure/risk' in order to guide the reader more/avoid confusion, or alternatively combine them. For example in the beginning you refer to temperature and air pollution as atmospheric conditions then later refer to them as risks. Thus, a smoother transition or immediately connecting the two is recommended (e.g. Extreme variation in atmospheric conditions such as temperature change and air pollutant concentrations pose important risks for health. Increased air pollution has been directly associated with increased mortality. Whilst non optimal temperatures (defined as xxx) has simultaneously been implicated with increased mortality etc..). I think it is a hard job to be navigating the language norms in between two fields so I applaud the effort for bridging the two fields of climate science and epidemiology together!

- Line 74 is a bit confusing does the 69% refer to being 69% of the 6.9 million or to environmental risk factors. Clarify.

- Line 85 Earth Systems Models (ESMS)

- Though the temperature has been examined in many investigations it has not been investigated many times in relation to health outcomes with air pollution simultaneously at the global level. In my paper reviewing current scenario literature and future health impacts we found only two papers at the global level examining air and climate simultaneously (however more exist examining specific individual countries) . If interested: [https://www.thelancet.com/journals/lanplh/article/PIIS2542-5196\(23\)00110-9/fulltext#seccestitle60](https://www.thelancet.com/journals/lanplh/article/PIIS2542-5196(23)00110-9/fulltext#seccestitle60) it could be a nice way to show the added value of this study.

Methods

-typo line 482 : post processed; 438: registration maybe send through a Grammarly checker to catch minor typos.

- In GBD 2019 there is not one TMREL used there is a range based on the lowest observed ranges in North America so check if it accurate to use just one TMREL and not a random draw of many TMREL values ranging from 2.4 to 5.9 as this is what I am seeing done in other papers.

- Really cool use of SEDAC data!

Results

- Figure 2 appears in the discussion but this is more a editing issue. Figures look good.

- Maybe add a mitigation scenario (as stated before) if time permits.

- Well done!

Discussion

- Perhaps since the supplementary analysis shows that more 'absolute number of' deaths are projected for cold deaths (despite the rate being lower than the heat non optimums) something could be added in the discussion about how climate mitigation also contributes to less extreme cold weather.

- Line 265 – 270 is a bit confusing.

- Might be interesting to mention that higher income countries may be more able to adapt to non-optimum temperatures/which sub-groups could be more impacted given unequal access or ability for adaptation (e.g. people who work outside).

Reviewer #2 (Remarks to the Author):

Dear Author,

I have thoroughly examined the manuscript titled "Atmospheric health burden across the century: the accelerating impact of temperature compared to pollution." The study employs model outputs from recent Intergovernmental Panel on Climate Change multi-institution simulations to evaluate prospective alterations in mortality linked to prolonged exposure to both non-optimal temperature and air pollution. While the manuscript showcases commendable quality, I would like to raise a few concerns that warrant consideration before contemplating its publication.

General Remarks:

- Please consider refining the title to make it more concise and focused on the main elements of the study.
- Please ensure consistency in the use of terminology and maintain a clear and coherent writing style throughout the section.
- Consider incorporating subheadings to enhance the overall structure and readability of the section.
- Encourage the integration of more recent references, if available, to strengthen the relevance and currency of the discussion.
- The methodology section should be moved below the introduction section

Abstract:

1. Explicitly state the research objectives in the abstract. What is the specific goal or goals of the study?

2. Some sentences are lengthy and could be broken down for better readability. Additionally, consider rephrasing certain sentences to improve flow and coherence.

3. The keywords effectively capture the main themes of the study. However, consider adding a few more specific keywords related to the methods used, such as "IPCC simulations" or "climate change scenarios," to enhance searchability.

Introduction:

1. It would be helpful to briefly define ESMs (Earth System Models) when first mentioned in the text to aid readers who may not be familiar with the term.

2. Consider providing a concise statement regarding the motivation behind choosing the SSP2-4.5 and SSP5-8.5 scenarios. This could include a brief explanation of their significance and relevance to the study.

3. The explanation of using PM2.5 as the focus for air pollution is justified, considering its substantial contribution to mortality. However, it may be beneficial to include a brief sentence explaining why PM2.5 is a key component in air pollution and its specific health implications.

4. The mention of downscaling and bias correction in Section 5 is noted. However, it would be helpful to provide a brief sentence explaining why these processes are necessary and their impact on the accuracy of the estimates.

5. More justifications and supporting references are required in the Introduction section. Some of the works that need to be cited include:

<https://doi.org/10.1007/s40572-020-00281-6>

<https://doi.org/10.1080/26896583.2023.2267332>

<https://doi.org/10.1016/B978-0-12-820730-7.00017-3>

Results

1. The statement "Significant attributable mortality from both risk factors is estimated in South and East Asia" could be enhanced by briefly discussing why these regions experience higher mortality, providing context for readers unfamiliar with the factors influencing mortality in these areas.
2. The use of Figures S1-S3 and Table S1-S3 is mentioned in relation to country-level mortality, but there's no reference to these figures and tables in the manuscript. Consider providing appropriate references or integrating key insights from these supplemental materials into the main text if relevant.
3. The substantial increase in global mortality by the end of the century is well-presented, and the figures effectively illustrate the projected changes. It might be helpful to include a brief discussion on the potential implications of such increases, emphasizing the significance of the findings.
4. The substantial increase in warm temperature attributable mortality is highlighted, and the regional distribution is discussed. It might be beneficial to briefly discuss potential societal and health implications of such increases, particularly in regions where the impact is most pronounced.

Discussion

1. It's valuable that the limitations of the study are discussed, particularly regarding the estimation of baseline mortality ratios. The consideration of potential offsets due to changing baseline mortality ratios and the decision to keep present-day values constant for the future are well-explained. This discussion adds nuance to the interpretation of the results.

We thank the reviewers for their comments. Here they are repeated (in bold) with our replies.

Reviewer # 1

General

- Very cool and interesting use of the most recent GBD data techniques and novel risk measures for impact assessment! Methods seem reproducible and in line with current climate modelling norms.

We thank the reviewer for the positive feedback.

- Be careful with the interpretation here when stating ‘non-optimal temperature as more important’ as built into the SSPs are measures that reduce air pollution, whilst built into the scenarios there is an assumption around temperature increase. Probably reducing climate change will also improve air quality. SSP5-8.5 is unlikely to occur and is more the ‘worst case scenario’.

The referee is right. We added the reference to Hausfather and Peters (2020), showing that SSP5-8.5 is highly unlikely to occur, while SSP2-4.5 is still a most probable future. We also now cite the work of Rao et al. (2017), where a detailed investigation of air pollution emissions in the SSP scenarios is present, also mentioning that “climate mitigation scenarios (e.g. 4.5 and 2.6 cases) result in most cases in co-benefits in terms of lower pollutant emissions than the baselines”, as also mentioned by the referee.

- In the body of the text you highlight the importance of air pollution policy but the title of the piece makes it seem that temperature and air pollution are pitted against each other. At the moment air pollution is a bigger share of burden of disease relative to temperature, thus I think there can also be a different way to frame the issue. I think that titling it “Atmospheric Health Burden Across the Century: the Accelerating Impact of Temperature “ is still a quite effective title. Also I think it’s a bit of a unfair comparison if I understand things correctly because the temperature mortality relationships are not taking into consideration adaptation measures while the PM2.5 calculations are, since SSP5 has stringent air pollution control and SSP2 has intermediate air pollution control. To demonstrate combined mitigation and air pollution benefits maybe an analysis using SSP1-2.6 could be used? That could reinforce your conclusion that effective climate change policies are also important

We agree with the referee that the SSP5-8.5 scenario is a somewhat unfair scenario, as it contains stringent air pollution control but almost no climate change policies. On the other side, SSP2-4.5 contains a climate mitigation scenario. Therefore, the comparison between the SSP5-8.5 and the SSP2-4.5 already shows the effect of climate policies, which could decrease significantly the non-optimal temperature-related mortality. These information have been added in the “Discussion” section. On the other side, we would like to mention that direct comparison between scenarios is not straightforward, as the population (and especially its age) changes significantly between the scenarios. As shown in the manuscript (lines 173-196), aging is a very important factor in mortality attribution calculations, and such difference could results in misinterpretation of the results. Therefore we refrain from adding additional scenarios, as doing so would likely require more discussion than allowed by the text length limit.

Abstract

- The concluding statement regarding ‘simultaneously implementing more effective climate change policies’ is very nice. I agree but not sure if the analysis demonstrates this since there is no explicit comparison of air pollution control scenarios to mitigation control scenarios.

We do agree with the referee that there is no explicit comparison of air pollution control scenarios to mitigation control scenarios. Nevertheless, we clearly show the effect of the no mitigation control scenario. Furthermore, even under scenario SSP2-4.5, which is still considered plausible (Hausfather and Peters, 2020), the change in non-optimal temperature mortality show a large increase, which should be considered when discussing the urgency of mitigation controls. We therefore prefer to keep the statement.

- Check whether or not British English or American English is used for the word ‘aging/ageing’ in Nature/the editor.

We thank the referee for the hint. Following Nature’s guidelines to author, American English should be used, and therefore we will correct the text to “aging” consistently.

- I think that the phrase including the word ‘potentially’ for SSP5-8.5 should be rewritten into something more moderate.

Based on the previous answers and on the unlikely of scenario SSP5-8.5, we decided to remove the references to SSP5-8.5 from the abstract, keeping the information only in the main text and mentioning this to be a very unlikely scenario.

- Specify briefly how this work either adds onto or reinforces other scientific findings if word limit permits. E.g. It is probably one of the few analyzing temperature and air pollution simultaneously at the global level , or that air quality improvements may be offset by non-optimal temperatures

We decide to highlight the fact that air quality improvements effort could be offset by non-optimal temperatures changes, as this fits well the final sentence of the abstract.

Introduction

- Be more specific in general when making broad claims and statements for example about ‘non-optimal temperatures’ as there are varying definitions which in some papers vary based on location in order to control slightly for local acclimatization/adaptation (e.g. Gasspirinis ‘Projections of temperature-related excess mortality under climate change scenarios’ 2017 article)

We do agree with the referee, and this is discussed in detail in the “Discussion” section.

- Choose one way to frame the ‘exposure/risk’ in order to guide the reader more/avoid confusion, or alternatively combine them. For example in the beginning you refer to temperature and air pollution as atmospheric conditions then later refer to them as risks. Thus, a smoother transition or immediately connecting the two is recommended (e.g. Extreme variation in atmospheric conditions such as temperature change and air pollutant concentrations pose important risks for health. Increased air pollution has been directly associated with increased mortality. Whilst non optimal temperatures

(defined as xxx) has simultaneously been implicated with increased mortality etc..). I think it is a hard job to be navigating the language norms in between two fields so I applaud the effort for bridging the two fields of climate science and epidemiology together!

We thank the referee for the nice suggestion. We reformulated the introduction to accommodate a better definition (and link) of exposure and risk.

- Line 74 is a bit confusing does the 69% refer to being 69% of the 6.9 million or to environmental risk factors. Clarify.

We agree that the sentence is not very clear. 69% refers to the contribution to the 6.5 million people per year. We have clarified in the revised version.

- Line 85 Earth Systems Models (ESMS)

This points was raised also by referee #2. We added a definition of Earth System Model, mentioning that ESMS are “global climate models with the added capability to explicitly represent biogeochemical processes” (Flato, 2011).

- Though the temperature has been examined in many investigations it has not been investigated many times in relation to health outcomes with air pollution simultaneously at the global level. In my paper reviewing current scenario literature and future health impacts we found only two papers at the global level examining air and climate simultaneously (however more exist examining specific individual countries) . If interested, (Weber et al., 2023) could be a nice way to show the added value of this study.

We thank the referee for pointing out the study. Nevertheless, from the work of Weber et al. (2023), we could find only the work of West et al. (2013) and Reis et al. (2022) who investigated the effect of climate change policies on air quality mortality. Therefore, no global study so far has investigated simultaneously the impact of non-optimal temperature and air pollution changes at global level. We will add the citation of Weber et al. (2023) in the introduction.

Methods

typo line 482 : post processed; 438: registration maybe send through a Grammarly checker to catch minor typos.

We thank the referee for the suggestion.

- In GBD 2019 there is not one TMREL used there is a range based on the lowest observed ranges in North America so check if it accurate to use just one TMREL and not a random draw of many TMREL values ranging from 2.4 to 5.9 as this is what I am seeing done in other papers.

We thank the referee for pointing this out. Indeed, the TMREL was based on an uniform distribution between 2.4 and 5.9 $\mu\text{g}/\text{m}^3$, following the GBD approach.

- Really cool use of SEDAC data!

Thanks for the nice comment.

Results

- **Figure 2 appears in the discussion but this is more a editing issue. Figures look good.**

We added the figures in the discussion so to easily follow them. The figure location will anyhow be changed if the paper is accepted.

Maybe add a mitigation scenario (as stated before) if time permits.

As mentioned before, we prefer to avoid adding an additional scenario, as this would require additional discussion on aging and population changes, which would not fit in the manuscript's text limitation.

Discussion

- **Perhaps since the supplementary analysis shows that more 'absolute number of' deaths are projected for cold deaths (despite the rate being lower than the heat non optimums) something could be added in the discussion about how climate mitigation also contributes to less extreme cold weather.**

This is partially discussed in lines 185-198. We have added here the reference to Rummukainen (2012), showing that climate change has increased the temperature variability and not only the average temperature.

- **Line 265 – 270 is a bit confusing.**

We have rephrased the sentence and made it simpler. The differences between our work and the work of Gasparrini et al. (2015) are due to the different relationship between temperature and relative risk. In the work of Gasparrini et al. (2015), this relationship is location-specific and includes socio-economic levels as well.

- **Might be interesting to mention that higher income countries may be more able to adapt to non-optimum temperatures/which sub-groups could be more impacted given unequal access or ability for adaptation (e.g. people who work outside).**

We agree with the referee that the impact could be different for different population sub-groups. We have added this information in the manuscript with a reference to Son et al. (2019), in which a systematic review on effect modifiers was conducted.

Reviewer # 2

Dear Author,

I have thoroughly examined the manuscript titled "Atmospheric health burden across the century: the accelerating impact of temperature compared to pollution." The study employs model outputs from recent Intergovernmental Panel on Climate Change multi-institution simulations to evaluate prospective alterations in mortality linked to prolonged exposure to both non-optimal temperature and air pollution. While the manuscript showcases commendable quality, I would like to raise a few concerns that warrant consideration before contemplating its publication.

We thank the referee for the constructive comments.

General remarks:

Please consider refining the title to make it more concise and focused on the main elements of the study.

Although we agree that the title could be improved, this comment disagrees with the suggestion of referee #1, who thinks “that titling it ‘Atmospheric Health Burden Across the Century: the Accelerating Impact of Temperature’ is still a quite effective title”. We therefore decided to keep the title, although we are open to suggestions also from the editor for a more effective title.

Please ensure consistency in the use of terminology and maintain a clear and coherent writing style throughout the section.

The text has been revised for consistency in the terminology and writing style.

Consider incorporating subheadings to enhance the overall structure and readability of the section.

We thank the referee for the suggestion. We have added subheadings in the “Results” and “Discussion” sections.

Encourage the integration of more recent references, if available, to strengthen the relevance and currency of the discussion.

We have added the references of Chen et al. (2020); Ayejoto et al. (2023); Singh et al. (2020), as suggested by the referee, as well as the one listed at the end of this document. Nicely, the review of Chen et al. (2020) shows that no work has been done before investigating simultaneously projected air pollution and non-optimal temperature-related mortality. Furthermore, this is the first work at global scale, as only analysis at urban or regional scale (e.g. Singh et al., 2020; Ayejoto et al., 2023) has been performed.

The methodology section should be moved below the introduction section.

Following the journal’s guidelines, we cannot move the “Methods” section into the main text. Firstly, we will exceed the number of words allowed in the main text, and secondly, the methodology section “appears in all online original research articles [...]”, and therefore should be located outside the main text.

Abstract:

1. Explicitly state the research objectives in the abstract. What is the specific goal or goals of the study?

The goal of the study is mentioned in the abstract : “to assess future changes in mortality attributable to long-term exposure to both non-optimal temperature and air pollution.” The abstract has been partially revised, so to give more emphasis to the research objective and also following the recommendation of referee #1.

2. Some sentences are lengthy and could be broken down for better readability. Additionally, consider rephrasing certain sentences to improve flow and coherence.

The abstract has been rephrased to improve its readability.

3. The keywords effectively capture the main themes of the study. However, consider adding a few more specific keywords related to the methods used, such as "IPCC simulations" or "climate change scenarios," to enhance searchability.

We have added the keywords suggested by the referee.

Introduction:

1. It would be helpful to briefly define ESMs (Earth System Models) when first mentioned in the text to aid readers who may not be familiar with the term.

We added that ESMs are "global climate models with the added capability to explicitly represent biogeochemical processes" (Flato, 2011).

2. Consider providing a concise statement regarding the motivation behind choosing the SSP2-4.5 and SSP5-8.5 scenarios. This could include a brief explanation of their significance and relevance to the study.

Clearly, scenario SSP2-4.5 has been chosen for its high probability to happen (Hausfather and Peters, 2020). On the other side, scenarios SSP5-8.5 represents the most (very unlikely) pessimistic scenario for climate change despite a very aggressive policy on air quality. The combination of the two give us a reasonable range of scenarios, without the need to include additional scenario. This information has been added to the Introduction section.

3. The explanation of using PM_{2.5} as the focus for air pollution is justified, considering its substantial contribution to mortality. However, it may be beneficial to include a brief sentence explaining why PM_{2.5} is a key component in air pollution and its specific health implications.

We have added the information that PM_{2.5} can pass through the respiratory barrier and enter the circulatory system, causing cardiovascular disease and other complications (Fang et al., 2013).

4. The mention of downscaling and bias correction in Section 5 is noted. However, it would be helpful to provide a brief sentence explaining why these processes are necessary and their impact on the accuracy of the estimates.

It is well known that Earth System Models have different accuracy in reproducing present day temperature and pollution (Turnock et al., 2020). Furthermore, their resolution is relatively coarse for health-related studies. Therefore, the model results have been downscaled so to increase their resolution, as well as bias corrected, so to have an accurate comparison with present day atmospheric status. We have added this information in the methodology, so to keep the main text within the limits.

5. More justifications and supporting references are required in the Introduction section. Some of the works that need to be cited include:

- <https://doi.org/10.1007/s40572-020-00281-6> ,
- <https://doi.org/10.1080/26896583.2023.2267332> ,
- <https://doi.org/10.1016/B978-0-12-820730-7.00017-3>.

We thank the referee for pointing out these references that we have overlooked. We have added them in the Introduction section.

Results

1. The statement "Significant attributable mortality from both risk factors is estimated in South and East Asia" could be enhanced by briefly discussing why these regions experience higher mortality, providing context for readers unfamiliar with the factors influencing mortality in these areas.

As shown in the methodology, the mortality is proportional to population number and age, as well as pollution/temperature exposure and baseline mortality ratio. For South and East Asia, the main driving term in the equation is the population number and the pollution exposure. In fact, while these regions contains 44% of the world population, they account for 46% and 61% for the attributable mortality to non-optimal temperature and pollution, respectively. We have added this information in the revised version.

2. The use of Figures S1-S3 and Table S1-S3 is mentioned in relation to country-level mortality, but there's no reference to these figures and tables in the manuscript. Consider providing appropriate references or integrating key insights from these supplemental materials into the main text if relevant.

The tables S1-S3 as well as the Figures S1-S3 are referred to in the text in lines 106-108. A detailed discussion on the country-level results are impossible in this manuscript due to the text limit. Nevertheless, we would like to keep the country-level information in the supplement, as this could be helpful for future investigation on these topics, as well as informative for readers worldwide.

3. The substantial increase in global mortality by the end of the century is well-presented, and the figures effectively illustrate the projected changes. It might be helpful to include a brief discussion on the potential implications of such increases, emphasizing the significance of the findings.

Following the suggestion of referee #1 we have added in the abstract and in the conclusions that the increase in non-optimal temperature exposure could potentially offset the gain from reduction in air pollution.

4. The substantial increase in warm temperature attributable mortality is highlighted, and the regional distribution is discussed. It might be beneficial to briefly discuss potential societal and health implications of such increases, particularly in regions where the impact is most pronounced.

The increase in warm temperature attributable mortality could have significant impact, especially in the vulnerable MENA region (Middle East and North Africa), further exacerbating the societal implications of climate change (e.g. migration and farmers' life quality) as described in Waha et al. (2017). We have added this sentence at the end of the results section.

Discussions

1. It's valuable that the limitations of the study are discussed, particularly regarding the estimation of baseline mortality ratios. The consideration of potential offsets due to changing baseline mortality ratios and the decision to keep present-day val-

ues constant for the future are well-explained. This discussion adds nuance to the interpretation of the results.

We thank the referee for the nice evaluation

References

- Ayejoto, D. A., Agbasi, J. C., Nwazelibe, V. E., Egbueri, J. C., and Alao, J. O. (2023). Understanding the connections between climate change, air pollution, and human health in africa: Insights from a literature review. *Journal of Environmental Science and Health, Part C*, 41(3-4):77–120.
- Chen, K., Vicedo-Cabrera, A. M., and Dubrow, R. (2020). Projections of ambient temperature- and air pollution-related mortality burden under combined climate change and population aging scenarios: a review. *Current Environmental Health Reports*, 7(3):243–255.
- Fang, Y., Mauzerall, D. L., Liu, J., Fiore, A. M., and Horowitz, L. W. (2013). Impacts of 21st century climate change on global air pollution-related premature mortality. *Climatic Change*, 121:239–253.
- Flato, G. M. (2011). Earth system models: an overview. *Wiley Interdisciplinary Reviews: Climate Change*, 2(6):783–800.
- Gasparrini, A., Guo, Y., Hashizume, M., Lavigne, E., Zanobetti, A., Schwartz, J., Tobias, A., Tong, S., Rocklöv, J., Forsberg, B., et al. (2015). Mortality risk attributable to high and low ambient temperature: a multicountry observational study. *The lancet*, 386(9991):369–375.
- Hausfather, Z. and Peters, G. (2020). Emissions – the ‘businessas usual’ story is misleading. *Nature*, 577:618–620.
- Rao, S., Klimont, Z., Smith, S. J., Van Dingenen, R., Dentener, F., Bouwman, L., Riahi, K., Amann, M., Bodirsky, B. L., van Vuuren, D. P., et al. (2017). Future air pollution in the shared socio-economic pathways. *Global Environmental Change*, 42:346–358.
- Reis, L. A., Drouet, L., and Tavoni, M. (2022). Internalising health-economic impacts of air pollution into climate policy: a global modelling study. *The Lancet Planetary Health*, 6(1):e40–e48.
- Rummukainen, M. (2012). Changes in climate and weather extremes in the 21st century. *Wiley Interdisciplinary Reviews: Climate Change*, 3(2):115–129.
- Singh, N., Singh, S., and Mall, R. (2020). Urban ecology and human health: implications of urban heat island, air pollution and climate change nexus. In *Urban ecology*, pages 317–334. Elsevier.
- Son, J.-Y., Liu, J. C., and Bell, M. L. (2019). Temperature-related mortality: a systematic review and investigation of effect modifiers. *Environmental Research Letters*, 14(7):073004.
- Turnock, S. T., Allen, R. J., Andrews, M., Bauer, S. E., Deushi, M., Emmons, L., Good, P., Horowitz, L., John, J. G., Michou, M., et al. (2020). Historical and future changes in air pollutants from cmip6 models. *Atmospheric Chemistry and Physics*, 20(23):14547–14579.

- Waha, K., Krumpalauer, L., Adams, S., Aich, V., Baarsch, F., Coumou, D., Fader, M., Hoff, H., Jobbins, G., Marcus, R., et al. (2017). Climate change impacts in the middle east and northern africa (mena) region and their implications for vulnerable population groups. *Regional Environmental Change*, 17:1623–1638.
- Weber, E., Downward, G. S., Ebi, K. L., Lucas, P. L., and van Vuuren, D. (2023). The use of environmental scenarios to project future health effects: a scoping review. *The Lancet Planetary Health*, 7(7):e611–e621.
- West, J. J., Smith, S. J., Silva, R. A., Naik, V., Zhang, Y., Adelman, Z., Fry, M. M., Anenberg, S., Horowitz, L. W., and Lamarque, J.-F. (2013). Co-benefits of mitigating global greenhouse gas emissions for future air quality and human health. *Nature climate change*, 3(10):885–889.

REVIEWERS' COMMENTS

Reviewer #1 (Remarks to the Author):

Dear authors,

Thank you for considering my previous feedback. I am glad you took the time to incorporate the remarks I made previously about SSP5-8.5 being highly unlikely, and for adding in the additional analysis of SSP2. I think the paper adequately addresses my concerns and I hope that the feedback helped in some way. I think the findings are very important at this moment and I wish you success in the final submission.

Warmly,

Reviewer #2 (Remarks to the Author):

Approved